# Non-Specific Signal Peptidase Processing of Extracellular Proteins in *Staphylococcus aureus* N315

**DOI:** 10.3390/proteomes11010008

**Published:** 2023-02-11

**Authors:** Santosh A. Misal, Shital D. Ovhal, Sujun Li, Jonathan A. Karty, Haixu Tang, Predrag Radivojac, James P. Reilly

**Affiliations:** 1Department of Chemistry, Indiana University, 800 E Kirkwood Avenue, Bloomington, IN 47405, USA; 2Luddy School of Informatics, Computing, and Engineering, Indiana University, 700 N. Woodlawn Avenue, Bloomington, IN 47408, USA; 3Khoury College of Computer Sciences, Northeastern University, 177 Huntington Avenue, Boston, MA 02115, USA

**Keywords:** *Staphylococcus aureus*, type I signal peptidase, N-terminal amidination, mass spectrometry

## Abstract

*Staphylococcus aureus* is one of the major community-acquired human pathogens, with growing multidrug-resistance, leading to a major threat of more prevalent infections to humans. A variety of virulence factors and toxic proteins are secreted during infection via the general secretory (Sec) pathway, which requires an N-terminal signal peptide to be cleaved from the N-terminus of the protein. This N-terminal signal peptide is recognized and processed by a type I signal peptidase (SPase). SPase-mediated signal peptide processing is the crucial step in the pathogenicity of *S. aureus*. In the present study, the SPase-mediated N-terminal protein processing and their cleavage specificity were evaluated using a combination of N-terminal amidination bottom-up and top-down proteomics-based mass spectrometry approaches. Secretory proteins were found to be cleaved by SPase, specifically and non-specifically, on both sides of the normal SPase cleavage site. The non-specific cleavages occur at the relatively smaller residues that are present next to the −1, +1, and +2 locations from the original SPase cleavage site to a lesser extent. Additional random cleavages at the middle and near the C-terminus of some protein sequences were also observed. This additional processing could be a part of some stress conditions and unknown signal peptidase mechanisms.

## 1. Introduction

Protein secretion is one of the essential cell functions in both prokaryotic and eukaryotic cells. Secreted proteins are translocated or transported from one location to the other, to perform specific functions outside of the cell. Many bacteria have dedicated protein secretion mechanisms and machinery to transport a wide variety of proteins and other biomolecules outside of the cell. Dedicated protein secretion machinery are generally found in pathogenic microorganisms, including Gram-positive and Gram-negative bacterial strains [1]. Secreted proteins in pathogenic bacteria often play significant roles in cell adhesion, cell anchoring, and the promotion of virulence, by enhancing their attachment to host cells. These pathogenic bacteria are also known to scavenge resources in an environmental niche, and they directly intoxicate target host cells and disrupt vital functions [2,3,4,5].

*Staphylococcus aureus* is one of the important Gram-positive pathogenic bacteria that occur in and on the human body. It causes a wide range of serious infections and some lead to life-threatening conditions, including bacteremia, pneumonia, endocarditis, osteomyelitis, toxic shock syndrome, and septicemia. *S. aureus* invades almost every tissue of the human body and secretes a variety of biomolecules, including cytotoxins, enterotoxins, proteases, lipolytic enzymes, and peptidoglycan hydrolases [6]. Secreted proteins are often toxic to the host cells, and they are generally known as virulence factors. *S. aureus* is noteworthy for the variety of virulence factors that it secretes and exports from its cell surface and eventually into host cells or tissue. The cytoplasmic membrane of the host cell is the main target where bacterial toxins pass through via pore formation. These molecules generally cause membrane damage and were previously categorized as receptor-mediated and receptor-independent toxins [1]. However, a few more membrane-damaging toxins bind and damage the host membrane non-specifically. Alpha-toxins are receptor-mediated toxins that lyse the red blood cells and leukocytes [7]. Such toxins, along with other proteolytic enzymes, are synthesized in the bacterial cytoplasm and are translocated outside of the cell via specific secretory pathways. The proteins that are to be exported outside the cell contain N-terminal leader peptide sequences, which are key determinants for particular secretory pathways [8,9,10]. Gram-positive bacteria, including *S. aureus*, have at least six different secretion pathways, consisting of the general secretory pathway (Sec-type), the twin-arginine (Tat) pathway, the pseudopilin-like (Com) pathway, ESAT-6, bacteriocin, and the Holins leader peptide pathway [11,12,13,14].

The Sec pathway was originally evolved to translocate the polypeptides across the plasma membrane utilizing an N-terminal signal peptide. A systematic mutational alteration in export machinery to study protein localization was first reported by Emr et al., in *E. coli* [15]. Generally, the majority of exported proteins are secreted via the general secretory (Sec) pathway after the N-terminal signal peptides are processed using type I and type II SPase (SPase I and II) [12,16]. In this pathway, the unfolded proteins are initially targeted to the membrane translocation machinery by specific export chaperones, and then the Sec machinery transports them across the membrane [17]. After translocation, the proteins undergo post-translational folding and chemical modifications [1]. Most of these proteins are translocated in their unfolded state, and the Sec pathway cleaves the typical N-terminal signal peptide during the process. The cleaved signal peptide sequence is divided into three parts, the N, H, and C domains. N is the N-terminus of the peptide, consisting of one or more positively charged residues, followed by a hydrophobic domain consisting of 10–20 hydrophobic amino acids. C is the C-terminal region that plays an important role in recognizing the SPase cleavage site [8,9]. The typical SPase cleavage occurs at the protein sequence of approximately 30 residues from the N-terminus of the protein, but its location varies from protein to protein. The determination of the exact SPase cleavage sites and their mechanisms was previously studied by utilizing genomics and proteomics methods [6,18,19,20].

In our previous study, we identified and characterized 59 secretory proteins in methicillin-resistant *S. aureus* COL via two-dimensional chromatography and mass spectrometry [6]. We also investigated many secretory proteins, including virulence factors, proteolytic enzymes, and peptidoglycans, in the extracellular milieu of the *S. aureus* COL, using different analytical techniques. In our recent work, we utilized the N-terminal chemical derivatization of protein/peptides with S-methyl thioacetimidate (SMTA) to discover a specific location of the SPase processed N-terminus of secreted proteins in *S. aureus* N315 (Figure 1) [21]. The major advantage of this method is to capture the SPase cleavages in the secreted proteins before enzymatic digestion. The SMTA reagent not only captures the endogenous cleavages, but it also improves the detectability of SMTA-modified peptides in the mass spectrometer. The N-terminus-modified peptides fragment differently in the high-collision dissociation that yields a series of b-ions. Typically, b_1_ ions are generally small and are not observed in HCD MS/MS spectra [22]. The SMTA reagent adds an amidino tag at the N-terminus that increases the mass of the b_1_ ion by 41.0256 Da, and improves the likelihood of observing b_1_ fragments. The b_1_ ion in the MS/MS spectrum definitively locates the labeling site and enhances the confidence of peptide identification. Approximately, 16 signal peptide cleavage sites in the secreted proteins were confidently determined [21,23]. Moreover, the N-termini of many secreted and some cytoplasmic proteins with and without initial methionine were also identified with the amidination bottom-up approach. To further explore the SPase processing and protein secretion, we utilized a similar approach and used protease inhibitors to stop unwanted protease cleavages after protein secretion. SPase activity was not inhibited by protease inhibitors. Interestingly, we discovered the non-specificity of the SPase, and also confidently identified many unexpected SPase cleavages in the middle and near the C-terminus of the secreted proteins. The cleavages identified using this approach were corroborated using top-down protein analysis. We have identified over 200 proteins in the extracellular milieu by using amidination and bottom-up approaches. The extracellular proteins contain virulence factors; proteolytic enzymes, peptidoglycans, and membrane-associated proteins. Most of the highly abundant proteins identified are reproduced in eight replicate experiments.

## 2. Materials and Methods

### 2.1. Materials

The *S. aureus* N315 strain was obtained from the Romesberg lab (The Scripps Research Institute, La Jolla, CA, USA). Brain heart infusion broth (BHI), LB broth, and tryptic soy broth (TSB) were purchased from Difco/BD biosciences. LCMS-grade acetonitrile and water were purchased from EMD Chemicals (Gibbstown, NJ, USA). Sequencing grade trypsin was purchased from Sigma Aldrich (St. Louis, MO, USA). Trichloroacetic acid, formic acid, and iodoacetamide were from Macron Fine chemicals. Dithiothreitol was purchased from Bio-Rad, USA.

### 2.2. Bacterial Growth and Culture Conditions

The stock culture of *S. aureus* N315 cells was streaked on a brain heart infusion (BHI) agar plate and grown overnight at 37 °C. Single colonies were selected, inoculated in 5 mL of sterile BHI broth (37 g/L), and incubated overnight at 37 °C in an orbital shaker. Then, 1 mL of this overnight culture was further inoculated in 100 mL of fresh BHI or tryptic soy broth (TSB) and grown for 6 h at 37 °C. The working culture was prepared by inoculating 100 mL of fresh sterile BHI or TSB broth with 1 mL of glycerol stock and incubating overnight at 37 °C in an orbital shaker at 250 rpm. A volume of 5 mL of this overnight culture was diluted to 500 mL with fresh BHI or TSB broth and allowed to grow for 6 h at 37 °C. To keep the SPase active, and to inhibit the other proteases, Roche cOmplete protease inhibitor cocktail tablets (1/50 mL) were added to the culture medium. The bacterial cell growth was monitored by measuring the absorbance at 660 nm using a UV-visible spectrophotometer, up to 0.8 to 1 optical density. Cells were pelleted via centrifugation at 8000 rpm for 10 min using the JA10 Beckman Coulter rotor. The cell pellet was stored at −80 °C for further study, and about 500 mL of the resultant supernatant was filtered through a 0.22 μm Millipore Stericup filter to remove any residual bacteria. Secreted proteins were precipitated in 10% TCA (*w*/*v*) overnight, with continuous stirring at 4 °C. Precipitated proteins were centrifuged at 15,000 RPM for 45 min, and the resulting pellet was washed with ice-cold acetone at least three times. The protein pellet was dried in a vacuum evaporator for 10 min and re-dissolved in 100 mM ammonium bicarbonate, pH 8, for chemical labeling and peptide analysis. Protein concentration was determined through a Bradford assay, using BSA as a standard.

### 2.3. Chemical Labeling of Extracellular Proteins with SMTA

SMTA was prepared as reported previously [24]. The TCA-precipitated protein pellet was dissolved in 100 mM ammonium bicarbonate, pH 8, with vigorous mixing and sonication. The protein solution was clarified via centrifugation at 15,000 RPM to remove the insoluble matter. An aliquot containing 600 µg protein was amidinated with 500 mM of SMTA for 2 h at room temperature. The pH of the reaction mixture was adjusted to 7.5–8 with 250 mM KOH. The reaction was stopped by adding 0.1% formic acid (FA). Approximately 300 μg of labeled protein sample was loaded on a strong cation exchange column (Tosoh Bioscience GmbH, TSKgel SP-NPR, 4.6 mm ID × 3.5 cm) for chromatography on a Waters alliance 2695 HPLC system. The strong cation exchange (SCX) column was pre-equilibrated with mobile phase A: 6 M urea and 20 mM glacial acetic acid, pH 5.0. After loading the protein sample on the column, excess unreacted SMTA was washed with mobile phase A for 20 min at a flow rate of 0.3 mL/min. Proteins were eluted with 0 to 90% B (6 M urea, 20 mM glacial acetic acid, and 0.5 M NaCl, pH 5.0) for a 120 min gradient at a flow rate of 0.3 mL/min, and the eluent was captured on pre-equilibrated 20 reverse phase C4 trapping columns (BioBasic 4 Javelin columns, 1.0 mm × 20 mm, Thermo Scientific) [25]. The C4 columns were desalted with 95% of 0.1 % FA in water and 5% ACN for 5 min each, and eluted with 90% ACN and 0.1% FA at a flow rate of 0.3 mL/min. The elution from 20 trapping C4 columns was collected in 20 fractions using a Waters fraction collector. The solvent from each fraction was evaporated to dryness in a vacuum evaporator, and the proteins were re-dissolved in 25 mM ammonium bicarbonate, pH 7.8. Protein disulfide bonds were reduced with 5 mM DTT, and alkylated with 5 mM iodoacetamide final concentration. Trypsin was added to each of the samples at a 1:50 ratio (trypsin: protein), and incubated for 18 h at 37 °C. The digestion was stopped using the addition of 0.1% FA, and the digests were centrifuged at 14,000 RPM for 10 min before loading onto the reverse phase C18 column.

A control experiment was performed to determine whether the amidination occurred on tryptic peptides via residual SMTA during trypsin digestion. About 300 μg of proteins were amidinated, separated, and fractionated, as described above. In each of the 20 fractions, 1 µg of Cytochrome C was added and digested with trypsin for 18 h at 37 °C. The digested fractions were treated the same way as SMTA-labeled samples.

### 2.4. Chemical Labeling of S. aureus Cell Lysate with SMTA

Cell pellets were washed twice with 20 mM HEPES, 100 mM NH_4_Cl, pH7.4; and dissolved in lysis buffer (20 mM HEPES, 100 mM NH_4_Cl, 10.5 mM Mg Acetate, 0.5 mM EDTA, 5 mM 2-mercaptoethanol, 3 mM phenylmethylsulfonyl fluoride (PMSF), and Roche cOmplete protease inhibitor tablets). The cells were lysed using Emulsifier EmulsiFlex-C3 at 25,000 PSI for 15 cycles. The lysate was collected and centrifuged at 15,000 RPM for 45 min in a Beckman Coulter JA-20 rotor. The protein concentration was determined via Bradford assay using BSA as a standard. The cell lysate was treated with 1 µg/µL of RNase A and 0.75 µg/µg of DNase for 30 min at 30 °C to remove the DNA and RNA. Approximately, 500 µg of proteins were amidinated with SMTA by following the protocol described above. Excess SMTA was removed by using an Amicon 3kDa MWCO centrifugal filter. Proteins were separated and fractionated by using SCX and C4 reverse-phase trapping columns, as described above. The protein solution was digested, and the resulting peptides were analyzed on a high-resolution Thermo Orbitrap XL and Fusion Lumos mass spectrometer.

### 2.5. LC-MS/MS Bottom-up Protein Analysis

Peptides were subjected to Eksigent nanoLC 2D and approximately 1 μg was loaded on a home-packed 15 cm long, 200 Å particle size, 5 μM C18 reverse phase column (Phenomenex, Torrance, CA, USA) with mobile phase A—0.1% FA in H_2_O, and eluted with a 100 min gradient of 0–90% mobile phase B—0.1% FA in ACN at a flow rate of 300 nL/min. Peptide solutions were electro-sprayed into the mass spectrometer, and the MS1 and MS2 data were recorded at resolutions of 60 K and 15 K, respectively. The MS1 intensity threshold was set to 1e3 to trigger MS2, and 40% HCD energy was used to fragment the precursor ions. Dynamic exclusion was enabled for 30 sec. The raw data were converted to Mascot generic format (.mgf) format using the MSConvert tool from ProteoWizard v3 (https://proteowizard.sourceforge.io/index.html) (accessed on 17 July 2015), and analyzed on Mascot (Matrix Science Inc.) and MSGF+ [26]. The proteome database of *S. aureus* N315 was downloaded from UniProt (https://www.uniprot.org/proteomes/UP000000751) (accessed on 17 July 2015). Peptides were searched with a target and decoy (reversed) proteome database. Semi-trypsin (non-specific cleavage at the N-terminal of the peptide) was selected as an enzyme with oxidation (M); amidination at the K- and the N-terminals of the peptide/protein were set as variable modifications. Precursor and fragment mass tolerance was set at 20 ppm and 0.02 Da, respectively. The identified peptides were filtered at ≥30, with a 1% False Discovery Rate (FDR). Another database search was performed on Mascot and MSGF+ with fixed lysine (K) amidination (+41.0265) and variable N-term amidination to find any database search artifacts (data not shown).

### 2.6. Orbitrap Fusion Lumos Tribrid Mass Spectrometry and Database Search

One out of eight sets of experiments were analyzed on an Orbitrap Fusion Lumos Tribrid mass spectrometer coupled with the nanoAcquity LC system. Approximately, 1 μg of peptides were loaded onto a nanoAcquity UPLC Symmetry C18 trap column (Waters) with 95% solvent A (0.1% FA in water (LCMS grade)) and 5% solvent B (0.1% FA in acetonitrile (LCMS grade)) using the nanoAcquity LC system at a flow rate of 300 nL/min. Peptides were separated on a C18 ACQUITY UPLC HSS T3 column (Waters) and eluted using a 60 min gradient from 3 to 48% of solvent B at a flow rate of 300 nL/min. The eluent from the C18 column was electrosprayed into the Orbitrap Lumos using a 1.8 kV spray voltage and a 260 °C ion transfer tube in positive mode. Peptides with the precursor mass range within *m*/*z* of 300 to 2000, with +2 to +7 charges, were selected for further fragmentation in MS2. The resolution for MS1 was set to 60,000 FWHM. The cycle time for MS1 scans was set to 3s. The automatic gain control (AGC) and the maximum injection time were set at 4e5 and 50 ms, respectively. The most intense precursor ions were selected and isolated using an ion trap with a peptide-like monoisotopic profile (MIPS) and an intensity threshold of 5e4, and fragmented with higher-energy collisional dissociation (HCD) (40% collision energy). The fragment ions (MS2) were analyzed in the orbitrap with a resolution of 30,000 FWHM at the auto-normal scan rate. The AGC target and the maximum injection time were set at 1e5 and 54 ms, respectively. The first mass of the MS2 scan was set to 90 *m*/*z* to detect b_1_ ions. Dynamic exclusion was enabled, and the precursor ion was fragmented twice and excluded for 30 s to avoid the repetitive acquisition of the same precursor ion having a similar *m*/*z* within ±10 ppm.

The mass spectrometry raw data were converted into .mgf format using the MSConvert tool, and the centroid peaks were created using the peak picking option. The resulting .mgf files were submitted to the Mascot/MSGF+ database search programs. The data were searched the same way as described in the previous section with semi-tryptic enzymatic cleavage. Mass tolerance for precursor and fragment ions was set to 5 ppm and 0.02 Da, respectively. Variable modifications were set as Oxidation (M) and Amidination (K and N-term of peptide/protein). The detected peptides on a Mascot within 1% FDR were considered as confident identifications. N-terminal amidinated peptides were manually checked for the presence of b_1_ ions.

### 2.7. LC-MS/MS Top-down Protein Analysis

The secreted proteins were isolated using TCA precipitation as described above. The TCA precipitated proteins were re-dissolved in 6 M urea and 2 M thiourea. The solution was mixed thoroughly with vortex and sonication, and clarified via centrifugation at 14,000× *g* for 10 min. Protein concentration was determined via a Bradford assay, using BSA as a standard. A total of 300 μg of proteins were loaded on a strong cation exchange chromatography column (TSKgel SP-NPR, 4.6 mm × 35 mm, Tosoh, Bioscience, Montgomeryville, PA, USA) using Waters Alliance 2695. The mobile phases A: 6 M urea, 20 mM glacial acetic acid, pH 5.0, and mobile phase B: 6 M urea, 20 mM glacial acetic acid, 0.5 M NaCl, pH 5.0 were used. Proteins were eluted with 0 to 90% B for a 120 min gradient, and the eluent was captured on 20 reverse phase C4 trapping columns (BioBasic 4 Javelin columns, 1.0 mm × 20 mm, Thermo Scientific) at a flow rate 0.3 mL/min. The C4 columns were desalted with another 95% mobile phase C containing 0.1% TFA in water, and 5% mobile phase D containing 0.1% TFA in acetonitrile (ACN) for 6 min each. Proteins were eluted with 5 to 90% of mobile phase D gradient, sent to another inline C4 analytical column (BioBasic 4, 5 μm × 0.5 mm, Thermo Scientific), and separated with a 90 min gradient using Agilent Infinity 1200. The eluent was electrosprayed into the Waters Synapt G1 HDMS mass spectrometer. Initially, the data were acquired only in MS1 mode, and analyzed on MassLynx 4.1. Another set of experiments was carried out with the top-down method, and the HDMS/MS data were acquired in positive ES+ mode using MassLynx v4.1. The scan range was selected from 250 to 2000 *m*/*z* for the MS survey and MS/MS. The MS/MS was triggered when the intensity threshold was raised above 50. The instrument parameters were set to a capillary voltage of 3 kV, a cone voltage of 30 V, a source temperature of 150 °C, a desolvation temperature of 400 °C, and a desolvation gas flow of 450 L/hr. The collision energy for low mass was set to 30, and for high mass, it was ramped up from 75 to 95 eV. All of the samples were run with lock mass spray acquired every 2 s during a 90 min survey scan. The raw data files were processed in MassLynx V4.1 using MaxEnt 1 and 3 to obtain the MS1 and MS2 peak lists. MaxEnt1 was used to deconvolute the mass of the protein, and MaxEnt3 was used to deconvolute the fragment ions of the intact protein mass. The top 50 peaks were selected from the MaxEnt3 deconvoluted spectrum and searched against the *S. aureus* N315 proteome database using the free edition of the Protein Analysis Worksheet Software (PAWS) (a freeware edition for Windows, ProteoMetrics, LLC, New York, NY, USA). Oxidation (M) and acetylation (N-term) were allowed as dynamic modifications. The location of the unknown signal peptide and the cleavage site prediction in the protein sequence was conducted with the web version of the SignalP server 5.0 (https://services.healthtech.dtu.dk/service.php?SignalP) (accessed on 14 April 2019) [27,28].

## 3. Results and Discussion

### 3.1. Non-Specific SPase Cleavages in the Secretory Proteins

To reveal the secreted proteome complexity, we analyzed the *S. aureus* culture supernatant through a combination of mass spectrometry-based proteomics methods. Previously, we identified and characterized 127 proteins in the *S. aureus* COL culture supernatant [6]. In present study, we identified multiple secretory proteins and their signal peptide cleavage sites. The SPase non-specific cleavages at the −1, +1, and +2 residues from the mature secreted protein are observed in a few secretory proteins that are listed in Table 1. The typical SPase cleavage site in the extracellular matrix-binding protein EbhB is (AHA^39^↓^40^AET) confirmed by amidinating the N-terminal of the peptide after SPase cleavage with a strong b_1_ ion in the MS/MS spectrum (Figure 2). In addition, we observed three other N-terminal-labeled peptides with cleavages (QAH^38^↓^39^AAET, AHAA^40^↓^41^ETN, AHAAE^41^↓^42^TNQ) at the −1, +1, and +2 residues after a typical SPase cleavage site in the same protein. Residues at −1 to +2 were A (alanine) and E (glutamic acid), which are relatively small and can easily be cleaved off during SPase processing. This indicates that the non-specific activity of the SPase is not unidirectional, but it is two-directional from the site of a normal cleavage site. The catalytic cleavage efficiency of SPase is more at the AHA^39^↓^40^AET site, and it decreases at the −1, +1, and +2 cleavage sites, as indicated by spectral counting, which is the total number of MS-MS spectra matching the observed peptide sequences in Table 1. A similar cleavage pattern was observed with Immunoglobulin G binding protein A, where one residue (A) is cleaved off from the −1 and +1 sites of the normal SPase cleavage site. In staphylocoagulase and staphylococcal complement inhibitor, SPase cleaves off the +1 residues I (Isoleucine) and S (Serine), respectively, from the normal signal peptide cleavage site.

Putative surface protein SA2285, a multilocation protein present in the cell wall, membrane, and extracellular milieu has a 1–50 residue signal peptide, and normal SPase cleavage (↓) occurs at AEA^50^↓^51^AEN that is confirmed through N-terminal amidination. The peptide AENNIENPTTLKDNVQSK with amidine label at the N-terminus and a strong b_1_ ion in the MS/MS was detected in 40 of 71 PSMs (Figure 3, Table 1). The 31 PSMs observed without an amidine label on N-terminus may arise due to the limited accessibility of the N-terminus to the reagent, or by having SMTA at too low a concentration in the reaction mixture. The presence of the b_1_ ion confirms that the N-terminus of the peptide is indeed labeled after SPase cleavage, and is not the artifact of a searching algorithm. Additionally, we also observed the peptides ENNIENPTTLKDNVQSK without A (alanine) at position +1, and we amidinated the N-terminus at E (Glutamic acid). The additional residue at +1 is cleaved off by SPase (AEAA^51^↓^52^ENN) with lesser PSM counts, as compared to the typical cleavage.

A similar cleavage pattern was observed in glutamyl endopeptidase where we observed the peptide SSKAMDNHPQQTQSSKQQTPK (cleavage at ANAL^30^↓^31^SSK) instead of the typical signal peptide cleavage site ANA^29^↓^30^LSSK. Interestingly, we did not observe the peptide starting with LSSK, suggesting that L is removed during the signal peptidase processing, or it might have yielded much lower abundant species than the observed peptide in the current experiment (Table 1 and Table 2, and Appendix A). The presence of two peptide populations in the same sample suggests that the SPase cleavage occurs at both places at the same time, with a lesser extent of one residue cleavage from +1 sites. This strongly suggests that the SPase cleavage is not always highly specific at the AXA cleavage site, and that it can occur at the −1, +1, or +2 residues from the normal signal peptide cleavage site. The non-specificity of the SPase could be alternative processing that occurred during the protein export, which cleaved off the small residues on either side of a typical cleavage site. This could also be the result of the modulation of SPase activity, linked with the increasing multi-drug resistance of *S. aureus*. Another possibility could be the result of a response to the stress condition.

### 3.2. Random Cleavages in the Secreted Proteins

The *S. aureus* secretory proteome is highly complex, due to processing with different secretory mechanisms. The secretory proteins and their proteoforms were poorly studied. The secreted proteome via the general secretory mechanism, and their signal peptides, were studied through the identification of the mature protein. The standard method for the determination of signal peptide cleavage sites by detecting the mature form of proteins using the intact mass by native mass spectrometry does not always specifically locate the cleavage sites. We recently described derivatizing the free amine ends of proteins/peptides with SMTA and analyzing them via high-resolution tandem mass spectrometry, to specifically locate the proteolytic cleavage sites [21]. The SMTA-modified peptides demonstrated more detectability, as compared to non-modified peptides and informative fragmentation patterns in HCD fragmentation [23]. By utilizing this approach, we confirmed several SPase cleavage sites, in agreement with those predicted for the common secretory proteins of *S. aureus*, in current and previous studies [21]. The SPase cleavage typically occurs after the Ala X Ala (AXA) motif in prokaryotes and eukaryotes [29,30]. The substrate specificity of SPase is determined by the presence of small and neutral side-chain residues at the −3 and −1 positions from the cleavage site [31,32]. Small non-polar amino acids were always found to be at the −1 and −3 positions of the signal peptide cleavage sites, and any residue is tolerated at the +1 site [33]. We observed that small residues that are present at the −1, +1, and +2 sites were occasionally cleaved off non-specifically during protein processing or translocation in *S. aureus* N315. To our knowledge, there is not any evidence of the non-specificity of SPase cleavage documented in *S. aureus* N315. However, there was the presence of the stable signal peptide in the extracellular matrix, and signal peptide cleavages at more than one site were reported previously [6,34].

Fibrinogen-binding protein SA1000 is an uncharacterized protein observed in the extracellular medium, with amidinated peptide being cleaved at the predicted signal peptide cleavage site. This offers additional confident experimental evidence to the SignalP 5.0 program signal peptide cleavage site prediction [27,28]. The MS/MS spectrum of the peptide ^30^QTKNVEAAKK, amidinated at the N-terminus with the presence of b_1_ ion, confirms the SPase cleavage site (Appendix A). Another cleavage was observed near the C-terminus of the protein sequence, with amidinated peptide at the N-terminus (^103^LKYNTLK, Appendix A). The MS/MS spectrum of the peptide amidination at the N-terminus demonstrates additional cleavage that likely occurred during SPase processing. The last 65 residues of this protein are predicted to be an extracellular fibrinogen binding C-terminal domain region [35]. Proteolytic processing near the N-terminus and the C-terminus might have some significance in the protein function and toxicity of the bacterial cell.

Cell division protein FtsZ is a cytoplasmic protein that is assembled at the inner cytoplasmic membrane and recruits other cell division proteins during cell division [36,37,38]. There is no previous experimental evidence that this protein is exported outside of the cell. We observed this protein in the extracellular medium, with one non-specific cleavage near the C-terminal domain (Table 2 and Appendix A). The peptide ^349^SNSSNAQATDSVSER from near the C-terminus of the protein was repeatedly observed amidinated in five experiments out of eight. The MS/MS spectrum in Appendix A shows excellent fragment ion matching with the peptide residues, and a strong b_1_ ion demonstrates the confident identification of the cleavage site. The C-terminal region is considered to be a highly flexible and disordered region in the protein, but essential for proper FtsZ assembly during cell division [37,39,40,41]. The presence of this protein in the extracellular medium might be the unexpected cell lysis during the growth.

We observed a total of 42 semi-tryptic peptides with N-terminal amidination and b_1_ ions in the MS/MS spectra. The MS/MS spectra of N-terminal amidinated peptides without b_1_ ions were treated as normal peptides, and they may be the artifacts of the database search program. All 42 novel cleavages in 34 proteins occurred in the middle of, or near the C-terminus of the protein sequence (Table 2 and Appendix A). Uncharacterized protein SA2202 is predicted to be localized in the cell membrane with the signal peptide. We observed the signal peptide cleavage SSK^24^↓^25^DKE, which was predicted by the SignalP5.0 program. Additionally, another N-terminus-labeled peptide was confidently identified (KPN^198^↓^199^AKI) about 60 residues upstream of the C-terminal of the protein sequence. The adapter protein MecA is a cytoplasmic protein that is involved in the targeting of unfolded and aggregated proteins to the ClpC ATPase [42]. We detected this protein in an extracellular medium with 60% sequence coverage (Appendix A). The cleavage (MSH^224^↓^225^NVT) was detected near the C-terminal domain. This region is probably interacting with ClpC protease or passing through the cell membrane, and it may become cleaved by extracellular proteases near the C-terminus.

Putative surface protein SA2285, probable transglycosylase IsaA, and extracellular matrix-binding protein EbhB are the most abundant proteins that are observed in the extracellular medium. These proteins are localized in the cell wall/cell membrane, as well as in the extracellular matrix. Standard SPase signal peptide cleavages were confirmed via N-terminal amidination for these three proteins. In addition to the typical SPase processing, we also observed a few other cleavages in the protein sequence at different locations (Table 2, and Appendix A). Their amidinated N-terminal peptides were observed with the presence of the b_1_ ion in the MS/MS spectra, suggesting the confident identification of a cleavage site. Although the SPase proteolytic preprotein processing occurs at the N-terminus of the mature proteins [12], it is also evident that the surface anchoring proteins are cleaved in more than one place during the preprotein processing and protein transport through the cell membrane [43]. Many surface proteins anchoring to the cell wall go through transpeptidase processing that requires the C-terminal sorting signal cleavage at the LPXTG motif [44]. Putative surface protein SA2285 is one of the abundant proteins that we observed; we confirmed the known and multiple unknown cleavages in top-down and bottom-up proteomics experiments [34]. This is a cell wall anchored protein with a predicted LPXTG sorting signal cleavage site, and it is similar to SasG, with seven G5 conserved domains. Interestingly, we did not observe the G5 domain peptides and LPXTG sorting cleavage in this protein in eight replicate experiments. There is no experimental evidence of this cleavage reported yet. Other cleavages at multiple locations could suggest that this protein is processed by other enzymes to respond different stress conditions during cell growth.

Lipase 2 and Thermonuclease are secreted proteins with known signal peptide cleavage sites. We did not observe their signal peptide cleavage in any replicate experiments. However, Thermonuclease propeptide cleavage at ANA^60^↓^61^SQT was observed in all replicate experiments. This cleavage resembles an AXA typical signal peptide cleavage pattern and is more likely cleaved by SPase during protein secretion [6]. No other non-specific cleavages were observed in Thermonuclease. In Lipase 2, the signal peptide (MLRGQEERKYSIRKYSIGVVSVLAATMFVVSSHEAQA^37^SEK^40^) containing the cleavage site at residue 37 was observed in an extracellular medium, suggesting that the mature protein was secreted without SPase processing. Additionally, a non-specific cleavage (YTG^670^↓^671^IIN) near to the C-terminus was detected confidently in five out of eight replicates. Three more proteins, UPF0223 protein SA0947, uncharacterized protein (A0A0H3JL00), and SA1273 protein, whose localization and function are not yet known, are also observed to be cleaved near the C-terminal domain. Other predicted cytoplasmic/cell membrane proteins cleaved near their C-terminal domain include 5-methyltetrahydropteroyltriglutamate–homocysteine methyltransferase, Transcriptional regulator CtsR, fructose-bisphosphate aldolase, and Alanine racemase 1. These proteins are more probably N-terminally anchored to the cell membrane. Lipoteichoic acid synthase involves lipoteichoic acid biosynthesis and cell wall biogenesis. We observe the two chains of this protein to be cleaved ALA^217^↓^218^SED, which corroborates with the predicted chain cleavage.

### 3.3. Non-Specific Cleavages Confirmed through Native and Top-down Proteomic Analysis

The standard signal peptide cleavages and non-specific cleavages in the secretory proteins identified using the N-terminal amidination bottom-up approach were further confirmed by the native and top-down proteomics approach. The top-down experiments without amidination gave additional evidence and confidence of the native cleavages in the secretory proteins.

We have identified 137 secretory proteins in the extracellular medium using native mass spectrometry. A total of 80 proteins were observed as fragments of various sizes, including some signal peptides (Appendix A). Intact protein masses were also observed with the removal of Methionine (M) or of one additional small residue from the N-terminus of the protein. Some proteins were observed, both with and without their N-terminal Met, including a few ribosomal proteins. The DNA-binding protein HU, and Major cold shock protein CspA, were observed abundantly in the extracellular medium. DNA binding protein HU is known to be an abundant cytosolic nucleoid-associated protein in most bacterial strains [45]. The presence of this protein in the extracellular matrix may have some role in cell adhesion and pathogenesis, as it was recently shown that some DNA-binding proteins were found in the *S. aureus* biofilm matrix [46]. Major cold shock protein CspA is another cytosolic protein that is abundantly expressed in stressed conditions. There are other intact proteins that were observed without initial M. Methionine cleavage by the methionine aminopeptidase is more efficient when M is followed by a small residue at the second position. Alanine (A), Serine (S), and Lysine (K) at the second position are more favorable for cleavage. Ribosomal proteins L13, L22, L24, and S6 were observed with initial intact Methionine. In order to confidently identify the proteins and to locate the specific SPase cleavage sites, the top-down protein analysis of extracellular proteins was carried out in parallel to the bottom-up amidination experiments. We confirmed over 30 protein fragments that were being cleaved from the middle of the protein sequence through top-down analysis (Appendix A).

In our previous publication, we reported the signal peptide cleavage and middle cleavage in Probable transglycosylase IsaA, via amidination bottom-up experiments [21]. Here, we report additional evidence of similar cleavages via both bottom-up and top-down mass spectrometry analysis of the extracellular proteins. In addition to the observed peptide followed by SPase cleavage, the stable signal peptide of Probable transglycosylase IsaA with initial Methionine was also repeatedly detected in the top-down experiments. In addition, the non-specific middle cleavage at AVS^130^↓^131^APT was also strongly observed in all bottom-up and top-down experiments (Appendix A). This confident identification of middle cleavage suggests that IsaA could be processed by SPase through an alternate processing mechanism during translocation.

The signal peptide cleavage and other middle cleavages from secreted uncharacterized protein A0A0H3JPH2 was repeatedly observed in amidination bottom-up experiments; those observations were corroborated with top-down analysis. This protein is predicted to be secreted outside of the cell with signal peptide cleavage at AEA^28^↓^29^ATV, and we confirmed this cleavage along with one additional cleavage in the signal peptide region at ILF^16^↓^17^GTL. The spectral counting suggests that the cleavage at ↓^29^ATV (120 PSMs) is more efficient than at ↓^17^GTL (20 PSMs). The lesser occurrence of ↓^17^GTL cleavage suggests that it might be an alternative cleavage that happens to perform different functions outside of the cell, along with another cleavage (SSF^69^↓^70^SGV) near the C-terminus of the protein. Together these three cleavages in the protein sequence at different locations could be a part of its unique molecular functions in *S. aureus* N315.

P61598 Putative surface protein SA2285 signal peptide cleavage at AEA^50^↓^51^AEN was observed in the amidination bottom-up experiment, along with other several middle cleavages. The middle cleavage at IYT^281^↓^282^SSM was observed in the bottom-up, and verified with top-down experiments (Appendix A). Other middle cleavages in the protein sequences that corroborate amidination and bottom-up experiments include LEK↓ELA (Valine-tRNA ligase valS), ILT^5^↓^6^IIL (A0A0H3JM99, Uncharacterized protein), SDW^20^↓^21^PEN, and ASK^181^↓^182^DVH, in different secretory proteins. The non-specific SPase cleavages observed in both bottom-up and top-down experiments in the presence of the protease inhibitors suggest that the signal peptidase processing is not inhibited [47].

The highly toxic and secreted *S. aureus* proteins could be potential drug targets to treat an infection, or vaccine candidates. SPase processed and preprocessed proteins, and their cleavage site information will be useful in designing potential biomarker discovery studies, as well as protein engineering studies. Potential protein markers could be used to make constructs and recombinant proteins. This cleavage site information needs to be further studied in detail for specific biological implications.

## 4. Conclusions

The bacterial cells were grown for 6 hr without any known stress conditions. About 42 new cleavage sites in 34 proteins occurred in the middle of, or near the C-terminus of the protein sequences. Most of these proteins are extracellular and are membrane-associated with some of the cytoplasmic proteins. The random cleavages in the secreted proteins are observed with and without the addition of protease inhibitors in the bacterial culture, suggesting that SPase processing may occur non-specifically in response to cell physiology, homeostasis, and stress conditions [47].

In conclusion, the data from the bottom-up amidination, and the native and top-down proteomic experiments suggest that the SPase processing is not highly specific at the signal peptide cleavage site. The extracellular matrix-binding protein EbhB, Immunoglobulin G binding protein A, and Immunoglobulin G binding protein A; and a few other proteins, as indicated in Table 1, show that the removal of one or two residues at either side occurs to a lesser extent, which could be linked to another specific molecular or cellular function. Furthermore, multiple known secretory proteins such as Probable transglycosylase IsaA, immunoglobulin-binding protein sbi, putative surface protein SA2285, immunoglobulin G-binding protein A, and uncharacterized proteins that were processed at the middle and near to the C-terminus of their sequence, could be the responses to different stress conditions during bacterial growth. Altogether, these data potentially explore the previously unknown function of SPase, and its cleavage non-specificity.

These SPase cleavages could be involved in the regulation of multiple virulence factors, including the production of toxins, and enzymes that contribute to the pathogenesis of *S. aureus* infections. Moreover, they significantly impact on the host immune response by modifying the activities of secreted proteins that are involved in inflammation, host defense, and other biological processes. The other significant biological significance of alternate cleavages could also be involved in controlling cell division and growth in *S. aureus*, helping to regulate the bacterial population size and virulence. SPase cleavages have also been shown to play a role in controlling the stability and homeostasis of the bacterial cell envelope, which is critical for the survival of the bacteria in the host.

## Figures and Tables

**Figure 1 proteomes-11-00008-f001:**
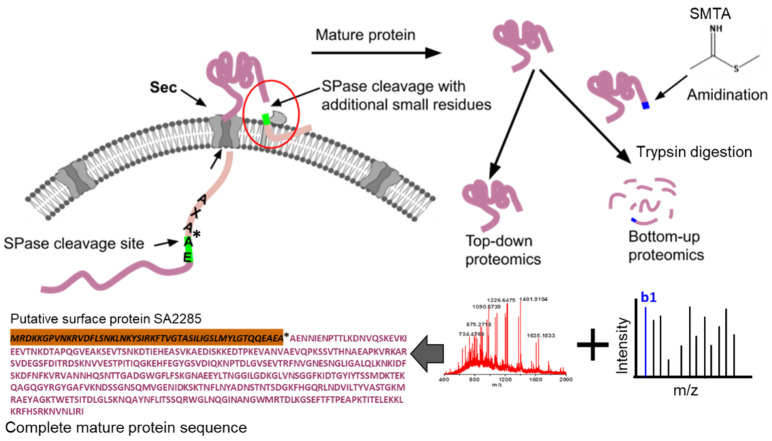
Overview of signal peptidase processing at non-specific amino acids in the protein sequence during Sec transport. The SPase cleavage site is precisely determined using the top-down and N-terminal amidination bottom-up proteomics methods.

**Figure 2 proteomes-11-00008-f002:**
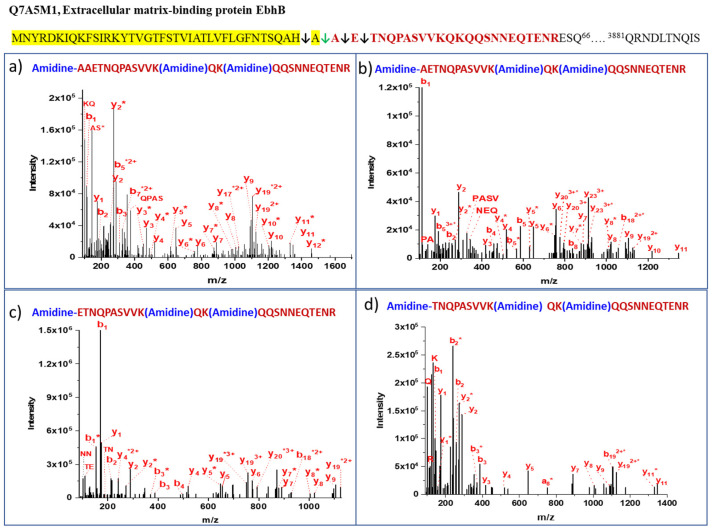
Nonspecific SPase processing. MS/MS spectra of four peptides from the extracellular matrix-binding protein EbhB demonstrate the SPase I cleavage specificity. The yellow highlighted sequence is the signal peptide. (**a**) A peptide AAETNQPASVVKQKQQSNNEQTENR contains one extra residue cleaved off from (−1) the signal peptide cleavage site (↓) at Ala39. (**b**) The peptide AETNQPASVVKQKQQSNNEQTENR is formed by normal SPase I cleavage at Ala40, and it is found to be more abundant. (**c**) In the ETNQPASVVKQKQQSNNEQTENR peptide, one more (+1) residue is cut off from the N-terminus during the SPase processing. (**d**)Two N-terminal residues (+2) are cut off from the TNQPASVVKQKQQSNNEQTENR peptide.

**Figure 3 proteomes-11-00008-f003:**
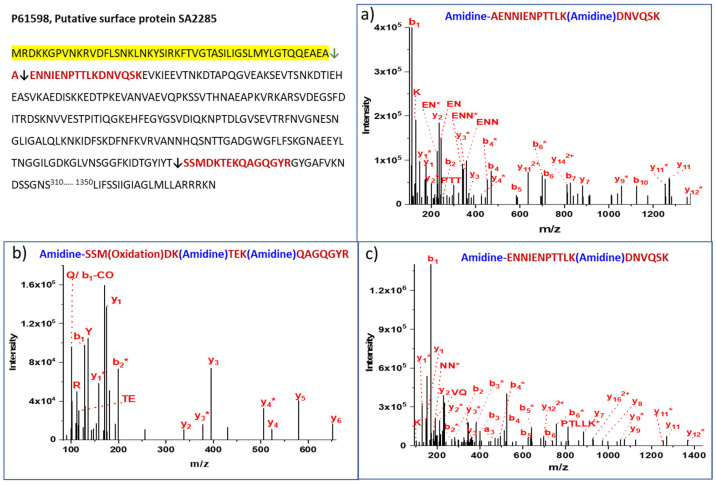
MS/MS spectra of the peptides from Putative surface protein SA2285. The top left panel shows the typical signal peptide cleavage occurs at AEA↓AEN and generates the peptide starting from AEN…EVK. The signal peptidase cleaves one more residue (+1) non-specifically from the mature protein (EAA↓ENN). The strong b_1_ ion in both cases confidently identified the modified peptides. (**a**) MS/MS spectrum indicating SPase cleavage at Ala50; (**b**) MS/MS spectrum indicating cleavage at Ser282; (**c**) MS/MS spectrum indicating cleavage at Ala51.

**Table 1 proteomes-11-00008-t001:** Non-specific SPase cleavages (the exact location of these cleavages in the full protein sequences are shown in Appendix A).

No.	Protein	Peptide	Cleavage Pattern	% b_1_ ion Abundance *	Total Peptides	Amidinated	Non-Amidinated	% Cleavage Frequency
1	Q7A5M1 Extracellular matrix-binding protein EbhB	AAETNQPASVVKQKQQSNNEQTENR	QAH↓AAET	46	3	3	0	10
2	Q7A5M1 Extracellular matrix-binding protein EbhB	AETNQPASVVKQKQQSNNEQTENR	AHA↓AET	100	21	21	0	68
3	Q7A5M1 Extracellular matrix-binding protein EbhB	ETNQPASVVKQKQQSNNEQTENR	HAA↓ETN	100	7	5	2	23
4	Q7A5M1 Extracellular matrix-binding protein EbhB	TNQPASVVKQKQQSNNEQTENR	AAE↓TNQ	35	2	2	0	6
5	P99134 Immunoglobulin G binding protein A	AAQHDEAQQNAFYQVLNMPNLNADQR	AAN↓AAQ		30	0	30	39
6	P99134 Immunoglobulin G binding protein A	AQHDEAQQNAFYQVLNMPNLNADQR	ANA↓AQH	47	41	38	3	53
7	P99134 Immunoglobulin G binding protein A	QHDEAQQNAFYQVLNMPNLNADQR	ANAA↓QHD	38	6	3	3	8
8	P61598 Putative surface protein SA2285	AENNIENPTTLKDNVQSK	AEA↓AEN	100	71	40	31	86
9	P61598 Putative surface protein SA2285	ENNIENPTTLKDNVQSK	EAA↓ENN	100	12	10	2	14
10	A0A0H3JNG8 Staphylocoagulase	IVTKDYSKESR	ADA↓IVT	100	17	17	0	59
11	A0A0H3JNG8 Staphylocoagulase	VTKDYSKESR	ADAI↓VTK	82	12	13	0	41
12	Q7A4V3 UPF0342 protein SA1663	VKANEESKKLFDEFR	AFAN↓VKA	16	3	3	0	43
13	Q7A4V3 UPF0342 protein SA1663	ANVKANEESKKLFDEFR	EAF↓ANV	20	4	4	0	57
14	Q99SU9 Staphylococcal complement inhibitor	STSLPTSNEYQNEKLANELK	AQA↓STS	100	24	20	4	77
15	Q99SU9 Staphylococcal complement inhibitor	TSLPTSNEYQNEK	AQAS↓TSL	4	7	4	3	23
16	Q7A6A6 Glutamyl endopeptidase	SSKAMDNHPQQTQSSKQQTPK	ANAL↓SSK	100	1	1	0	100
17	A0A0H3JNR9 Uncharacterized protein	SETNQKVSTNQESK	AEA↓SET	45	20	18	2	100

* % b_1_ ion abundance, relative to the most intense ion in the spectrum.

**Table 2 proteomes-11-00008-t002:** N-terminal amidinated peptides observed from the middle of the protein sequence, and they do not necessarily follow the AXA cleavage pattern.

No.	Protein	Peptide	Location in Protein	Cleavage Pattern	Subcellular Location	% b_1_ Ion Abundance *	Total PSMs	Amidinated	Non-Amidinated
1	A0A0H3JPH2 Uncharacterized protein	GTLLGVTYK	N-term	FVSSCIASTILF^16^↓^17^GTL	Extracellular	50	20	20	0
2	A0A0H3JPH2 Uncharacterized protein	TILFGTLLGVTYK	N-term	KKFVSSCIAS^12^↓^13^TIL	Extracellular	100	2	2	0
3	A0A0H3JM99 SA1477 Uncharacterized protein	IILIALLVILLFRVGLSILR	N-term	MSILT^5^↓^6^IIL	Predicted/transmembrane	50	3	2	1
4	P65986 DNA repair protein RecO	DVHAVILSNK	Middle	DGAISRQEASK^181^↓^182^DVH	Cytoplasm	20	2	2	0
5	P99160 Probable transglycosylase IsaA	APTYHNYSTSTTSSSVR	Middle	SSNSNVEAVS^130^↓^131^APT	Extracellular	100	47	28	19
6	P68800 Fibrinogen-binding protein	IVEYNDGTFKYQSR	Middle	EKKPVSINHN^46^↓^47^IVE	Extracellular	100	9	4	5
7	P65289 Lipase 1	NPNIVYKTYTGEATHK	Middle	ATDLNRKTSL^548^↓^549^NPN	Extracellular	80	9	9	0
8	Q99RL2 Immunoglobulin-binding protein sbi	LKGITEEQR	Middle	QQKAFYQVLH^60^↓^61^LKG	Extracellular	30	54	39	15
9	Q7A6A6 Glutamyl endopeptidase	LKPLEQR	Middle	QTPKIKKGGN^57^↓^58^LKP	Extracellular	40	24	24	0
10	A0A0H3JP98 SA0743 protein	APSKKPTTPTTYTETTTQVPMPTVER	Middle	DNENDRQLVVS^344^↓^345^APS	Extracellular	100	7	7	0
11	P61598 Putative surface protein SA2285	SSMDKTEKQAGQGYR	Middle	FKIDTGYIYT^281^↓^282^SSM	Extracellular/Cell wall	40	14	10	4
12	P61598 Putative surface protein SA2285	SVDIQKNPTDLGVSEVTR	Middle	QGKEHFEGYG^178^↓^179S^VD	Extracellular/Cell wall	100	97	16	81
13	P61598 Putative surface protein SA2285	EVTSNKDTIEHEASVK	Middle	TAPQGVEAKS^88^↓^89^EVT	Extracellular/Cell wall	30	3	3	0
14	P99134 Immunoglobulin G-binding protein A	IKPGQELVVDKK	Middle	ADNKLADKNM^390^↓^391^IKP	Extracellular/Cell wall	30	3	2	1
15	Q7A6P2 Thermonuclease	SQTDNGVNR	Middle	VSSLSSSANA^60^↓^61^SQT	Extracellular	100	22	22	0
16	Q7A6U1 Lipoteichoic acid synthase	SEDDLTKVLNYTKQR	Middle	IENNQQKALA^217^↓^218^SED	Cell membrane/Extracellular	100	4	4	0
17	A0A0H3JKR2 Penicillin binding protein 2 prime	EKLYDKKLQHEDGYR	Middle	DDAVIGKKGL^283^↓^284^EKL	Membrane	20	7	4	3
18	A0A0H3JTB6 Uncharacterized protein	SEVNENVEKQNHFKHR	Middle	DILNIHTAKA^67^↓^68^SEV	Predicted membrane	20	3	3	0
19	A0A0H3JNE5 SA0173 protein	ANTLVYR	Middle	GTEQMLGMF^1266^↓^1267^ANT	predicted/not known	y^n−1^	9	9	0
20	P64416 histidine ammonia-lyase	LQVNLIR	Middle	SDVRIDQYNE^74^ ↓^75^LQV	Cytoplasm/membrane	20	520	432	88
21	A0A0H3JQ77 Penicillin-binding protein 3	GETMVDEPLHFQGGLTKR	Middle	AGYQNKAIKV^410^↓^411^GET	Predicted/transmembrane	60	3	3	0
22	A0A0H3JKY5 SA1224 protein	LKILSGELDSQTGHVSLGKNER	Middle	GANGAGKSTF^43^↓^44^LKI	Predicted/ABC transporter	30	2	2	0
23	Q7A600 Probable dual-specificity RNA methyltransferase RlmN	STLGGLK	Middle	GCRIGCTFCA^135^↓^136^STL	Cytoplasm	10	6	6	0
24	P99135 Phosphoglycerate kinase	IKDLKEGDVLLVENTR	Middle	ETRGEKLEAA^103^↓^104^IKD	Cytoplasm	30	6	5	1
25	A0A0H3JLW4 Uncharacterized protein	PENIVEKYQYQDFDDMFKHYQQLINQCKVQFDNYVTGK	Middle	ADYEGWWLFSDW^20^↓^21^PEN	Predicted	100	3	2	1
26	A0A0H3JNR9 Uncharactrized protein	SINNKFINFEER	Middle	HTSVKGKVAL^221^↓^222^SIN	Predicted	20	107	27	80
27	P99134 Immunoglobulin G binding protein A	GEENPFIGTTVFGGLSLALGAALLAGR	Near C-term	LPET^419^↓^420^GEE	Extracellular	100	123	123	0
28	P99108 Cell division protein Ftsz	SNSSNAQATDSVSER	Near C-term	SNATSKDESFT^349^↓^350^SNS	Cytoplasm	90	28	28	0
29	A0A0H3JNV0 SA2202 protein	AKIKAIKGNAEQSR	Near C-term	LSYLDYKKQKPN^199^↓^200^AKI	Predicted membrane/Signal	30	18	15	3
30	Q7A7P2 Lipase 2	IINDLLR	Near C-term	RKGAELANFYTG^670^↓^671^IIN	Extracellular	20	9	9	0
31	A0A0H3JMK9 SA1273 protein	ALSAGQR	Near C-term	YQSVENVVENID^222^↓^223^ALS	not Known	100	4	4	0
32	Q7A423 Staphylococcal secretory antigen ssaA2	GPGVVTSR	Near C-term	NGSVRVSEMNYGY^245^↓^246^GPG	Extracellular	100	3	3	0
332	A0A0H3JK15 Uncharacterized protein	SKVAGNFGYIEKGK	Near C-term	YVGKAVTHTEY^131^↓^132^SKV	Predicted/Signal	10	13	11	2
34	A0A0H3JKR2 Penicillin binding protein 2 prime	LKMKQGETGR	Near C-term	YANLIGKSGTAE^602^↓^603^LKM	Predicted/transmembrane/signal	30	8	7	1
35	A0A0H3JM43 Uncharacterized protein	GMIMATINSKSEGMTEWER	Near C-term	TINKNYADDQTYYLS^407^↓^408^GMI	Predicted/membrane	100	2	2	0
36	A0A0H3JTW9 Cell division protein FtsL	SLENDNVKVVR	Near C-term	IYEKAKKQGM^115^↓^116^SLE	Cell membrane	10	2	2	0
37	P68800 Fibrinogen-binding protein	LKQGLVR	Near C-term	LQERIDNV^158^↓^159^LKQ	Extracellular	20	2	2	0
38	A0A0H3JKR2 Penicillin binding protein 2	VNKTHKEDIYR	Near C-term	LTDGMQQV^578^↓^579^VNK	Predicted/transmembrane	100	6	6	0
39	A0A0H3JPQ1 SA1000 Fibrinogen-binding protein	LKYNTLK	Near C-term	SYEKKKLQRQIDLV↓^103^LKY	Predicted Extracellular	20	32	32	0
40	A0A0H3JPH2 Uncharacterized protein	SGVARPGVQSKASAPK	Near C-term	STAVGKYSSF^69^↓^70^SGV	Extracellular	20	6	1	5
41	P60432 Ribosomal protein L2	SVMNPNDHPHGGGEGR	Near C-term	SRWKGIRPTVRG^222^↓^223^SVMNPN	Cytoplasm	10	623	623	0
42	P99152 Elongation factor Tu	APGSITPHTEFKAEVYVLSKDEGGR	Near C-term	GVAREDVQRGQVLA^294^↓^295^APG	Cytoplasm	100	6	6	0
43	P60185 Adapter protein MecA	NVTAQVR	Near C-term	YLNDYAKIIMSH^224^↓^225^NVT	Cytoplasm	100	21	21	0

* b_1_ ion abundance relative to the most intense ion in the spectrum. In some cases where b_1_ ion is not observed, the y^n−1^ is listed.

## Data Availability

The data presented in this study are available in the article, or in the Appendix A.

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
