# Peer review of "Non-Specific Signal Peptidase Processing of Extracellular Proteins in *Staphylococcus aureus* N315"

_proteomes, 2023, doi:10.3390/proteomes11010008_

Round 1
Reviewer 1 Report
The work of Misal et al. contains the identification of novel specific and non-specific SPase cleavage sites for S. aureus. Their method for protein N-terminus isolation and facilitated fragmentation in MS proved to be highly appropriate for this research question.
I have only minor points that should be addressed before publication, mostly pertaining to method description:
L135: to my understanding, the idea of adding the protease inhibitor was avoiding any further cleavage/degradation of secreted proteins by secreted proteases to conserve SPase cleavage. Please improve clarity here, because you intend to keep SPase activity, or not?
L139: about how many mL of supernatant was filtered? Add info
L155: add info: column dimension, product type for SCX, flow rate of LC, T of column
L160: add info: product type, flow rate, T, dimensions for C4 columns
L162: 20 fractions in total or one fraction per C4 column?
L185: C18 and not C4 column for proteins? Why choose C18?
L190: C18 particles type and manufacturer
L192: LC flow rate, T
L204: peptides were filtered with a>= 30 ??? What do you mean here?
L248: flow rate, T
L251: flow rate, T
L268: is there a publication reference for your software? Otherwise, your home-built program should be made available as supplementary material.
L320: instead?
L499: there seems to be something wrong with this sentence, please check and revise
Author Response
Referee 1:
The work of Misal et al. contains the identification of novel specific and non-specific SPase cleavage sites for S. aureus. Their method for protein N-terminus isolation and facilitated fragmentation in MS proved to be highly appropriate for this research question.
Author’s Response: We would like to thank the reviewer and appreciate the encouraging remarks.
I have only minor points that should be addressed before publication, mostly pertaining to method description:
L135: to my understanding, the idea of adding the protease inhibitor was avoiding any further cleavage/degradation of secreted proteins by secreted proteases to conserve SPase cleavage. Please improve clarity here, because you intend to keep SPase activity, or not?
Author’s Response: We thank the reviewer for pointing this out. Yes, the idea was to inhibit the other proteases and keep the SPase active. We now made it clear in the revised manuscript.
L139: about how many mL of supernatant was filtered? Add info
Author’s Response: About 500ml of supernatant was filtered. This info has been added to the revised manuscript.
L155: add info: column dimension, product type for SCX, flow rate of LC, T of column
Author’s Response: Added info in the revised manuscript.
L160: add info: product type, flow rate, T, dimensions for C4 columns
Author’s Response: Added info in the revised manuscript.
L162: 20 fractions in total or one fraction per C4 column?
Author’s Response: One fraction per C4 trapping column. A total of 20 C4 columns were used to collect 20 fractions.
L185: C18 and not C4 column for proteins? Why choose C18?
Author’s Response: C4 column for proteins. Corrected the typo in the manuscript.
L190: C18 particles type and manufacturer
Author’s Response: Added info in the revised manuscript.
L192: LC flow rate, T
Author’s Response: Added info in the revised manuscript.
L204: peptides were filtered with a>= 30 ??? What do you mean here?
Author’s Response: Clarified in the revised manuscript.
L248: flow rate, T
Author’s Response: Added info in the revised manuscript.
L251: flow rate, T
Author’s Response: Added info in the revised manuscript.
L268: is there a publication reference for your software? Otherwise, your home-built program should be made available as supplementary material.
Author’s Response: We used the customized free edition of Protein Analysis Worksheet Software (PAWS) (a freeware edition, ProteoMetrics, LLC, New York, NY) for the top-down analysis. Added this info in the revised manuscript.
L320: instead
Author’s Response: Corrected in the revised version of the manuscript.
L499: there seems to be something wrong with this sentence, please check and revise
Author’s Response: Revised the sentence for more clarity.
Reviewer 2 Report
The authors have carried out a nice and rather complete characterization of the processing of extracellular proteins in S. aureus. The experimental procedures are described very clearly and with great detail. Their results support the conclusions extracted from the experiments. There are, however, some questions that the authors should clarify.
There is no question about the heterogeneity in the cleavage position at the typical SPase cleavage site. However, the random cleavages are also assigned to SPase activity. It might be the case but, can the authors rule out the possibility of other protease activity? There might be some residual protease activity that could be responsible for at least some of the observed cleavages. Thus, the authors should provide some additional control or, at least, comment on the evidence leading to conclude that SPase is the only active protease present.
The authors comment that the efficiency toward de "canonical" cleavage site is higher than toward the other non-specific sites. Is it posible to extract some quantitative information regarding the specificity of the protease? At least in term of frequencies of specific vs non-specific cleavages. This could be an interesting information to understand and to characterize this protein activity.
The authors do not comment at all about the relevance of their findings in a biological context. The potential implications of the non-specific protein processing of proteins could be discussed in the context of the bacterial biology and in relation with its pathogenic activity, as it is mentioned in the introduction.
Author Response
Referee 2:
The authors have carried out a nice and rather complete characterization of the processing of extracellular proteins in S. aureus. The experimental procedures are described very clearly and with great detail. Their results support the conclusions extracted from the experiments. There are, however, some questions that the authors should clarify.
Author’s Response: We would like to thank the reviewer and appreciate the comments.
There is no question about the heterogeneity in the cleavage position at the typical SPase cleavage site. However, the random cleavages are also assigned to SPase activity. It might be the case but, can the authors rule out the possibility of other protease activity? There might be some residual protease activity that could be responsible for at least some of the observed cleavages. Thus, the authors should provide some additional control or, at least, comment on the evidence leading to conclude that SPase is the only active protease present.
Author’s Response: We thank the reviewer for raising this issue. Initially, we suspect there could be some residual proteases active in the S. aureus secretome and potentially contribute to additional cleavages in secreted proteins. To address this issue, we added a protease inhibitor cocktail in the cell culture medium and sample preparation steps to prevent other protease cleavages and keep the SPase active.
The authors comment that the efficiency toward de "canonical" cleavage site is higher than toward the other non-specific sites. Is it posible to extract some quantitative information regarding the specificity of the protease? At least in term of frequencies of specific vs non-specific cleavages. This could be an interesting information to understand and to characterize this protein activity.
Author’s Response: We thank the reviewer for this comment. We agree that this is a piece of really interesting information and needs to be added to the manuscript. We have calculated the relative frequencies of some cleavages mentioned in table 1 in the revised manuscript.
The authors do not comment at all about the relevance of their findings in a biological context. The potential implications of the non-specific protein processing of proteins could be discussed in the context of the bacterial biology and in relation with its pathogenic activity, as it is mentioned in the introduction.
Author’s Response: We thank the reviewer for pointing this out. The relevance of this study is added in the discussion and conclusion section of the manuscript. However, we think that these observation needs to be further studied for specific biological and clinical implications.